# Replica molding-based nanopatterning of tribocharge on elastomer with application to electrohydrodynamic nanolithography

Qiang Li [1], Akshit Peer [1,2,3], In Ho Cho[4], Rana Biswas[1,2,3,5] & Jaeyoun Kim [1,2,3]

Replica molding often induces tribocharge on elastomers. To date, this phenomenon has been studied only on untextured elastomer surfaces even though replica molding is an effective method for their nanotexturing. Here we show that on elastomer surfaces nanotextured through replica molding the induced tribocharge also becomes patterned at nanoscale in close correlation with the nanotexture. By applying Kelvin probe microscopy, electrohydrodynamic lithography, and electrostatic analysis to our model nanostructure, poly (dimethylsiloxane) nanocup arrays replicated from a polycarbonate nanocone array, we reveal that the induced tribocharge is highly localized within the nanocup, especially around its rim. Through finite element analysis, we also find that the rim sustains the strongest friction during the demolding process. From these findings, we identify the demolding-induced friction as the main factor governing the tribocharge's nanoscale distribution pattern. By incorporating the resulting annular tribocharge into electrohydrodynamic lithography, we also accomplish facile realization of nanovolcanos with 10 nm-scale craters.

[1] Department of Electrical and Computer Engineering, Iowa State University, Ames, Iowa 50011, USA. [2] Ames Laboratory, Iowa State University, Ames, Iowa 50011, USA. [3] Microelectronics Research Center, Iowa State University, Ames, Iowa 50011, USA. [4] Department of Civil, Construction and Environmental Engineering, Iowa State University, Ames, Iowa 50011, USA. [5] Department of Physics and Astronomy, Iowa State University, Ames, Iowa 50011, USA. Correspondence and requests for materials should be addressed to R.B. (email: biswasr@iastate.edu) or to J.K. (email: plasmon@iastate.edu)

Tribocharging of elastomer surfaces due to their electrical[1,2] or frictional[3–7] contact with other materials has been attracting substantial interest, with the resulting tribocharges already playing crucial roles in energy harvesting[8–10], mass spectrometry[11], and electronics[12,13]. They are strongly suggested to arise from the transfer of electrons or ions between material surfaces[1–7]. Recently, a similar tribocharging has also been observed on the surface of the elastomer poly(dimethylsiloxane) (PDMS) as the result of replica molding. The ensuing studies revealed that the level of tribocharging is strong enough to influence some microfluidic functionalities, such as channel electrophoresis[14–16]. So far, however, this replica molding-induced tribocharging phenomenon has been studied only on flat, untextured elastomer surfaces. It is rather ironic since replica molding is the primary method for surface texturing of the PDMS. Questions regarding how those textures affect the tribocharge's generation and distribution patterns, especially at nanoscale, have been left unanswered to date.

Here we attempt to answer the questions through a multi-physical investigation employing replica molded PDMS nanostructures. Since the resulting tribocharge distribution is below the range of direct imaging, we adopt indirect approaches which pair experimental techniques, such as scanning Kelvin probe microscopy (SKPM) and electrohydrodynamic lithography (EHDL), with iterative numerical modeling. In addition, we also model the replica molding process from the mechanical point of view. Based on the findings from the investigations, we identify the frictional stress, induced by the demolding action, as the main factor governing the tribocharge's nanoscale distribution pattern. This work also establishes a useful application for the resulting ring-shaped tribocharge by configuring it to enable EHDL to build nanovolcanos with 10 nm-scale nanocraters, a center-dimpled 3D nanostructure.

## Results

**Multi-physical investigation strategy**. As our model nanostructure, we adopt arrayed PDMS nanocups replicated from a polycarbonate (PC) nanocone array (Fig. 1a). We first image the tribocharge using SKPM. A recent application of SKPM to a flat PDMS surface, detached from a flat PC surface, revealed that the tribocharge distribution was an ensemble of negatively and positively charged nanodomains, with the overall polarity determined ultimately by the net charge[5]. In our case, SKPM of the PDMS nanocup arrays reveal that the tribocharge is highly localized inside the nanocups, making its distribution pattern accurately match the nanocup's array pattern. To find the tribocharge's distribution pattern inside the nanocup, we adopt two additional approaches.

First, we reconstruct the charge distribution through iterative electrostatic modeling which adjusts the model charge distribution until the computed electric potential agrees well with the experimental SKPM result. For our nanocup structure, a ring charge distribution around the rim produces the best match.

Second, we utilize EHDL in which liquid-phase polymer is attracted by spatially modulated electric fields to form out-of-plane structures[17–21]. We employ the tribocharged nanocups as the source of the spatially modulated electric fields (Fig. 1b). Then, we reconstruct the charge distribution through an iterative numerical simulation, which adjusts the model charge distribution until the simulation result agrees well with the experimental EHDL result. This tribocharge-enabled EHDL produced very unusual nanovolcanos with 10 nm-scale nanocraters, indicating that the polymer is attracted toward the nanocups' rims preferentially. In the numerical simulation, a ring charge distribution around the rim again produces the best agreement,

corroborating both the electrostatic modeling and the EHDL results.

Then, we perform an independent finite element analysis to find the origin of the ring shape in the charge distribution. It shows that the demolding process produces the maximum level of PDMS-PC friction around the rim area, successfully bridging mechanical phenomena with electrostatic observations.

**Scanning Kelvin probe microscopy of nanopatterned tribocharge**. Figure 1a shows the process to induce nanopatterned tribocharges on PDMS through replica molding. Using a PC mold patterned with a 750 nm-pitch nanocone array, we obtained a matching array of PDMS nanocups. The surface topography, examined with atomic force microscopy (AFM) in the tapping mode and scanning electron microscopy, are shown in Fig. 2a and Supplementary Fig. 1, respectively. The average depth $d$ was 153 ± 13 (s.d.) nm. To elucidate the polarity and the distribution pattern of the tribocharges on the replica molded PDMS surface, the surface potential was also measured through SKPM and plotted in Fig. 2b. More details on the AFM and SKPM procedures can be found in Methods.

From the comparison of the scans in Fig. 2a, b, it is evident that the positions of the negative potential wells closely match those of the nanocups' apertures. The surface topography and potential profiles shown in Fig. 2c, superimposed for facile comparison, further confirm their close correlation. Since the work function difference between the PDMS surface and the AFM probe is almost the same across the scanning area, the wells in the surface potential are induced mainly by the tribocharges[22]. It also indicates that the PDMS surface was negatively charged, which agrees well with the negative tribocharging of PDMS by PC reported by Baytekin et al.[5]. Interestingly, the surface potential exhibits peaks near the center of the nanocups, which yields valuable information on the charge distribution within the nanocups.

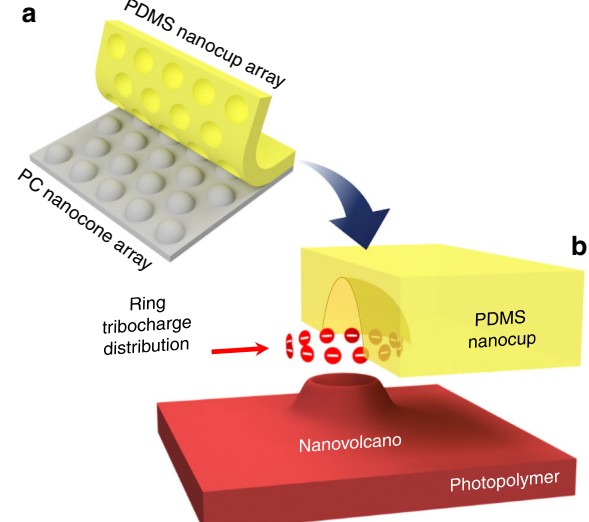

**Fig. 1** Replica molding-based tribocharging and its use in EHDL. **a** After being replica molded from a nanotextured polycarbonate (PC) mold, the elastomer replica's surface acquires tribocharges distributed in close correlation with the nanotexture. **b** The resulting electric field can subsequently shape the photopolymer at nanoscale through EHDL. In this work, the PDMS nanocup, replicated from a PC nanocone, acquires a nanoring-shaped tribocharge which shapes the photopolymer into a nanovolcano

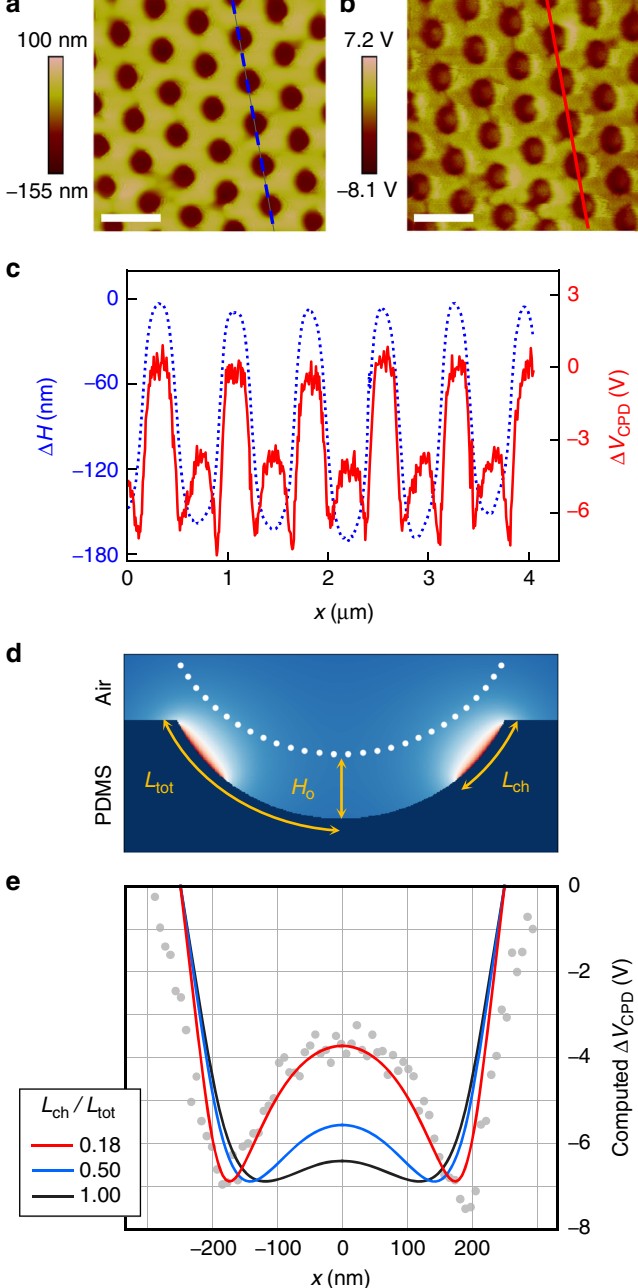

**Fig. 2** SKPM-based imaging and analysis of tribocharge distribution. **a** AFM image of the PDMS nanocup array's surface topography. **b** SKPM image of the surface potential $V_{CPD}$ at the same spot. (Scale bars: 1 μm) **c** Superimposed cross-sectional profiles of the surface topography and potential along the scan lines in **a**, **b**. The pattern overlap clearly indicates that the inner cavity of the nanocup is negatively charged. **d** A schematic diagram of the surface potential computation setup. $L_{tot}$ and $L_{ch}$ represent the arc lengths measured from the nanocup's rim to the bottom and the end point of the surface charge distribution, respectively. $H_o$ is the vertical gap maintained between the probe tips and the PDMS surface. The white dots represent the probing points for the surface potential measurement and evaluation. **e** The computed surface potentials for different charge distributions. They clearly show that the center peak rises within the potential well as the charge distribution becomes concentrated around the rim. In contrast, a dome charge ($L_{ch} = L_{tot}$) produces negligible center peak. The gray dots represent the experimental data in **c** within the 1.2 μm < $x$ < 1.8 μm range

**Electrostatic modeling of surface potential**. To extract more information from the SKPM results, we performed iterative electrostatic modeling which reconstructs the charge distribution by repeatedly adjusting the model charge distribution until the resulting electric potential exhibits a good agreement with the experimental measurement. Among the salient features of the SKPM result in Fig. 2b, c, of special concern was the peak inside the potential well. As shown in the charge distribution models and the corresponding electric potential computation results shown in Fig. 2d, e, such a center peak appears when a ring or annular strip-shaped charge distribution is dipped or penetrated by an AFM probe's tip and it becomes increasingly higher as the charge distribution becomes more concentrated around the nanocup's rim, reducing $L_{ch}$. In contrast, the peak becomes much lower in the case of a half-dome charge distribution ($L_{ch} = 0.5L_{tot}$) and almost disappears in a uniform dome charge distribution ($L_{ch} = L_{tot}$). Jacobs et al.[1] observed 'dip-in-the-peak' potential profiles, the inverse of our 'peak-in-the-well' profile, from their positive ring charges.

In Fig. 2c, the average ratio between the center peak height and the potential well depth was ~0.46 with the average potential well depth at 6.9 ± 0.7 (s.d.) V. As shown in Fig. 2e, the best match was obtained when the tribocharge was configured to form a ~55 nm-wide annular strip ($L_{ch} = 0.18L_{tot}$) around the rim. Under the assumption that the tribocharge is distributed in a bipolar mosaic form[5,23–27] with the overall polarity determined by the net charge, the corresponding net surface charge density is approximately $-9.9$ mC m$^{-2}$ or 0.6 net negative elementary charges per 10 nm$^2$, which is in order-of-magnitude agreement with the result reported by Baytekin et al.[5] (1 net negative elementary charge per 10 nm$^2$) for the same material combination (PDMS-PC)[5]. The fact that the potential stayed below the rim level throughout the PDMS nanocup's cavity strongly suggests that any portion of the PDMS nanocup not covered by the negative charge was uncharged or positively charged at a negligibly low charge density. Either way, our model of negative ring charge prevails. Details of the electrostatic model and charge distribution reconstruction can be found in Methods.

In SKPM-based analysis, however, the possibility that the observed surface potential pattern was a spurious projection of the surface topography is not entirely zero[28]. To test the ring charge hypothesis further, we performed EHDL of polymer using the tribocharged PDMS nanocup array as the source of the spatially modulated electric field and then reconstruct the tribocharge distribution through iterative numerical simulations.

**Surface pre-texturing for tribocharge-enabled EHDL**. In EHDL, liquid-phase polymer becomes polarized and attracted by spatially modulated electric fields and forms out-of-plane structures upon solidification[17–21]. Therefore, the gap between the source of the electric field and the polymer surface is one of the most important factors in EHDL. Conventional EHDL utilizes a patterned electrode as the source of the electric field and separately prepared dielectric thin film stripes as the spacers[17,29]. Here we utilized the tribocharged PDMS nanocups (Fig. 3a) as the source of the electric field. To place a gap between them and the polymer surface, we selected a photopolymer, which undergoes low but definite volume shrinkage upon exposure to UV irradiation[30], as the EHDL's target material and then textured the surface with a spatially modulated UV beam. The recesses in the resulting texture provide the gaps.

Specifically, we spin-coated UV-curable photopolymer NOA73 (Norland Inc.) into a thin film on a Si-substrate, and exposed it to a UV-two-beam interference pattern (Fig. 3b). Then the NOA73 surface became sinusoidally textured due to the local

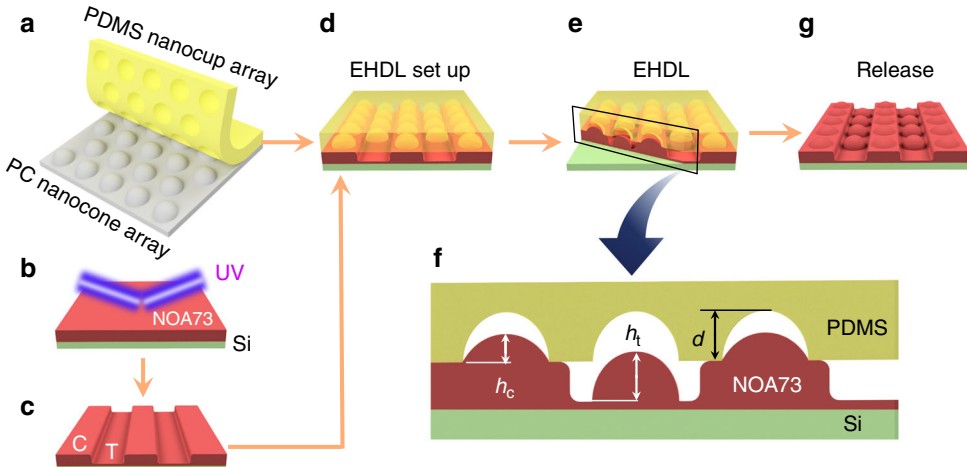

**Fig. 3** Fabrication steps for tribocharge-enabled EHDL of photopolymer. **a** Liquid-phase PDMS is poured onto the PC mold textured with a 2D triangular nanocone array. After thermal curing, the PDMS replica, textured with a nanocup array, is peeled off. Its surface becomes selectively tribocharged during this demolding process. **b** A UV-curable photopolymer (NOA73) is spin-coated on a silicon substrate and exposed to a UV-two-beam interference pattern. **c** The NOA73 thin film is textured sinusoidally with well-defined crest (C) and trough (T) areas due to local volume shrinkage. **d** The tribocharged PDMS nanocup array is placed on the sinusoidally textured NOA73 film. **e** NOA73 in the trough region is attracted upward by the spatially modulated electric fields originated from the tribocharges and undergoes EHDL. NOA73 on the crest experiences forces from both the capillary action and Coulomb attraction. **f** The cross-sectional profile defines the heights of the nanostructures in the crest ($h_c$) and trough ($h_t$) areas along with $d$, the nanocup depth. **g** The final UV-induced solidification of NOA73 and removal of the PDMS nanocup array completes the tribocharge-enabled EHDL of NOA73

volume shrinkage (Fig. 3c and Supplementary Fig. 2). More details on the sinusoidal texturing of NOA73 can be found in Methods. Note that even though the NOA73 thin film's inner volume becomes well cured by the UV exposure[31], a thin layer at its top surface remains fluidic and, hence, available for EHDL due to the oxygen-induced inhibition of photopolymerization[32–35]. When the tribocharged PDMS nanocup array was placed on the pre-textured NOA73 film (Fig. 3d), the troughs of the sinusoidal texture provide periodic recesses in which the NOA73 surface is vertically separated from the tribocharges by a submicron-scale gap.

Providing vertical separation through UV-induced texturing of the target material itself, rather than by adding heterogeneous spacers[17,29], leads to an additional merit. As illustrated in Fig. 3e, f, the crest portion of the sinusoidally textured NOA73 is in direct contact with the tribocharged PDMS nanocups and, hence, experiences both capillary action and tribocharge's Coulombic attraction. On the other hand, the trough portion, which is vertically separated from the tribocharged PDMS surface, experiences only the Coulombic attraction. This fact will prove useful in analyzing the EHDL results to corroborate the ring charge hypothesis.

**Tribocharge-enabled EHDL.** Upon completion of the photopolymer surface pre-texturing, we carried out the EHDL process. As shown in Fig. 3d–g, we placed the tribocharged PDMS nanocup array on the sinusoidally textured NOA73 thin film, left it for a preset period of time, and then applied the final UV irradiation to fix the final shape. The completely cured NOA73 film was peeled off from the PDMS surface and then examined by AFM. More details can be found in Methods.

Three different UV doses, 1.2, 1.8, and 3.6 J cm$^{-2}$, were used for the two-beam interference to produce different gap widths between the tribocharge and the NOA73 surface. AFM scans of the resulting three samples, to be referred to as Samples A, B, and C, are shown in Fig. 4. They reveal the impact of the UV dose on the final EHDL result. The scans from Samples A and B, shown as Fig. 4a, d, respectively, indicate that the EHDL process generated

nanocones arrayed on the top of the sinusoidally textured NOA73 surfaces, at locations matching those of the PDMS nanocups. The absence of parasitic protrusions on the NOA73 surface between the nanocones indicates that the flat, interstitial area between the nanocups' apertures hosted little or no net tribocharge. The nanocone array (~750 nm in pitch) and the sinusoidal texture (~2.6 μm in pitch) jointly constitute a two-level hierarchy which will be useful for many applications, such as superhydrophobic surfaces[36,37].

The trough nanocones, however, cannot be unambiguously attributed to EHDL yet. Given the high-level flexibility of PDMS[38], it is possible for the PDMS nanocup array to collapse down to the sinusoidally textured NOA73 surface, make a conformal contact with it, and produce the nanocones through capillary filling of the nanocups with the liquid-phase NOA73, rather than through EHDL. We, however, reject the conjecture based on the observation that the heights of the nanocones on the NOA73 crests ($h_c$ ~ 25 nm as shown in Scan 3 of Fig. 4f) and troughs ($h_t$ ~ 70 nm as shown in Scan 1 of Fig. 4f) are very different while the capillary filling-induced nanocones must exhibit similar heights. Moreover, the height of the crest nanocones is not just different from that of the trough nanocones but actually shorter. It is almost counterintuitive given the fact that the crests of the NOA73 texture corresponds to the destructive portion of the UV-two-beam interference pattern, which leaves NOA73 more fluidic and deformable. On the other hand, the trough portion of the NOA73 texture corresponds to the constructive part which cures NOA73 more intensely. Yet, the NOA73 in the trough resulted in higher nanocones. Based on these observations, we reject the conjecture of collapsed PDMS and attribute the trough nanocones unambiguously to the tribocharge-enabled EHDL.

**Underfilled crest nanocone as ring charge evidence.** The crest nanocones are more intriguing since their height is less than the depth of the PDMS nanocup ($d$ ~ 153 nm). It indicates that NOA73 failed to fill the nanocup completely. It was surprising since the time required for NOA73 to fill the PDMS nanocup

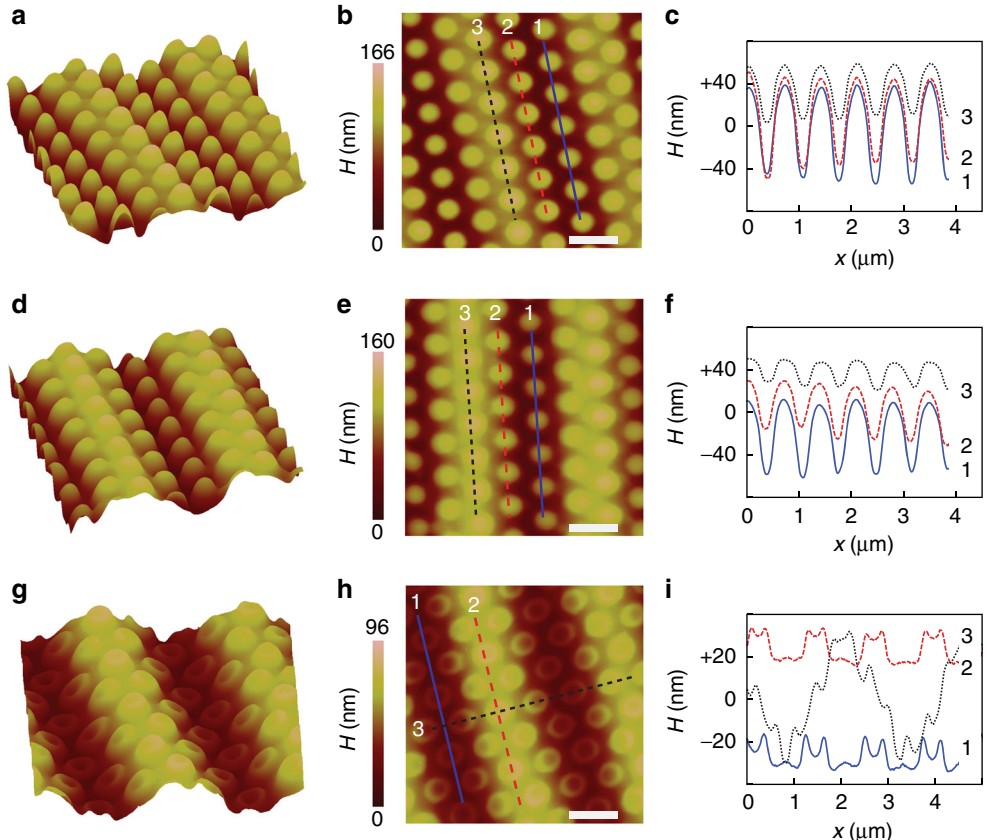

**Fig. 4** EHDL-generated nanocones and nanovolcanos. AFM scans of EHDL results obtained with the UV exposure dose of the two-beam interference lithography set to **a–c** 1.2 J cm$^{-2}$, **d–f** 1.8 J cm$^{-2}$, and **g–i** 3.6 J cm$^{-2}$. The first and second columns show the final textures in the bird's eye and top views, respectively. The third column shows their cross-sectional profiles along the lines in the second column. While the low dose, narrow-gap EHDL produced nanocone array as shown in the first two rows, the high dose, wide-gap EHDL resulted in a nanovolcano array as shown in the third row. (Scale bars: 1 μm)

through capillary action is <1 s according to[39,40]

$$t = \frac{2\,\mu\,d^2}{R\,\gamma\,\cos\theta} \qquad (1)$$

where $\mu$ is the viscosity of NOA73, $d$ is the PDMS nanocup depth, $R$ is the hydraulic radius of the nanocup, $\gamma$ is the surface tension of NOA73, and $\theta$ is the contact angle between NOA73 and PDMS. In our experiments, we maintained the contact between PDMS and NOA73 for at least 2 min. Yet, the filling was incomplete. By assuming that the tribocharges were distributed only around the nanocup's rim, we can explain this underfilling as the result of the attraction from the tribocharges which pulls down NOA73 toward the rim, counteracting the capillary flow toward the inner cavity[41].

**Nanovolcano formation as ring charge evidence**. The ring charge hypothesis can be further corroborated by the very unusual nanovolcano structures (Fig. 4g, h, i) produced by the tribocharge-enabled EHDL with the UV dose increased to 3.6 J cm$^{-2}$ (Sample C). Their biggest distinction from the nanocone structure is the nanocrater with 10 nm-scale height. The formation of the nanocrater indicates that NOA73 was attracted more strongly toward the rim of the nanocup's aperture than its center. If the tribocharges were distributed only along the nanocup's rim, they can attract the photopolymer in that fashion, as shown schematically in Fig. 1b. Under such a charge distribution, the nanocones in the troughs shown in samples A and B (Fig. 4a, d) can be interpreted as the result of the nanocrater's fusion at the

center of the nanocup due to the lower UV dose, which renders NOA73 more fluidic and dispersive.

Note that the height profiles in Fig. 4 could give the wrong impression that the sinusoidal texture is deeper in Fig. 4c than in Fig. 4i even though the former sustained a lower UV dose and, consequently, smaller shrinkage and shallower texturing. It can be explained by the fact that the upward deformation of photopolymer in both EHDL and capillary filling requires additional photopolymer. Therefore the nanocones in the trough in Fig. 4c achieved their height by lowering the bottom level around them, thus generating the illusion of a deeper trough.

To further corroborate the ring charge hypothesis, we proceeded to reconstruct the tribocharge distribution through iterative numerical simulations in which the model charge configuration was adjusted until a good agreement was reached between the experimental and simulation results. The two-dimensional model of the experimental setup is shown in Fig. 5a. The simulation is based on Eq. (2) which describes the nonlinear electrohydrodynamic interaction between the electric field and incompressible Newtonian fluid as[42–44]

$$\frac{\partial h}{\partial t} = \frac{\partial}{\partial x}\left(\frac{h^3}{3\mu}\cdot\frac{\partial P}{\partial x}\right) \qquad (2)$$

where $x$ is the lateral coordinate, $h(x, t)$ the height of the polymer surface in $y$-direction, $\mu$ the viscosity, and $t$ the time. $P$ is the pressure acting on the polymer surface and typically includes three components: the Maxwell stress, the Laplace pressure, and the disjoining pressure. They result from the Coulombic attraction, the interfacial tension, and the van der Waals

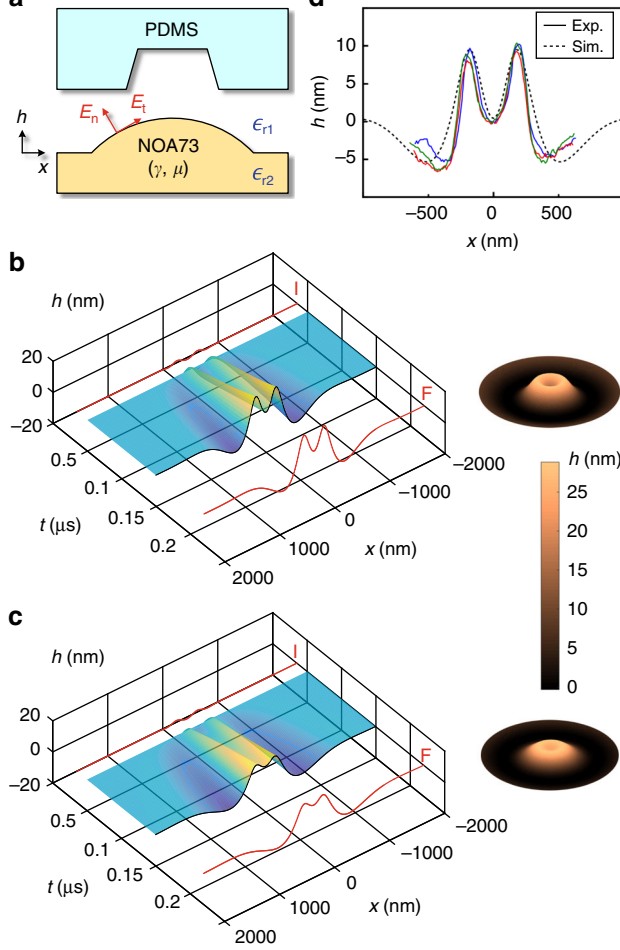

**Fig. 5** Numerical modeling of the EHDL process. **a** The 2D model for the numerical EHDL simulation. **b** The simulated evolution of the nanovolcano structure. The inset shows the revolved version of the final profile (marked as "F"). **c** The simulation result obtained after lowering the viscosity of NOA73. The nanocrater in **b** merged at the center to transform the nanovolcano into a nanocone. **d** The simulation (dotted line) and experimental (solid lines) results exhibit good agreements

interaction between the polymer and the electrode surfaces, respectively. Since the disjoining force becomes significant only when the polymer gets very close to the electrode, which is not the case in our EHDL, it is excluded from the simulation. Regarding the Maxwell stress, conventional EHDL simulations often include only the vertical, *y*-directed electric field[43]. Since our tribocharge-enabled EHDL setup utilizes non-uniform, highly localized charge distributions, we considered both normal and tangential electric fields at every point on the polymer surface. The overall pressure term becomes[45]

$$P = \gamma \cdot \frac{\partial^2 h}{\partial x^2} + \frac{\epsilon_0}{2}\left(\epsilon_{r1}^2 E_n^2 \cdot \left(\frac{1}{\epsilon_{r1}} - \frac{1}{\epsilon_{r2}}\right) + E_t^2 \cdot (\epsilon_{r2} - \epsilon_{r1})\right) \quad (3)$$

where the first and the second terms are the Laplace pressure and the Maxwell stress, respectively, $\gamma$ the interfacial tension of the polymer, $E_n(E_t)$ the strength of the electric field normal (tangential) to the polymer surface, $\epsilon_{r1,r2}$ the relative permittivity of the material, and $\epsilon_0$ the electric permittivity in vacuum.

We solved the governing equation numerically by integrating it over time *t*. The parameters were set to the values that are either measured or obtained from the literature. In particular, $\mu$ and $\gamma$ of

NOA73 were set to 130 cps and 0.04 N m$^{-1}$ (see Ref. 46). The absolute value of the surface charge density and its distribution shape were varied iteratively until a good match between the experimental and numerical results was obtained. Details can be found in Methods. Figure 5b shows the nanovolcano formation as a function of time.

Again, the best agreement between the simulation and experimental results was achieved when the tribocharge distribution was set to the form of a ring around the rim of the PDMS nanocup. Figure 5b clearly shows that the nanovolcano initially appears as an annular ridge induced by the ring charge (marked as "I"), becomes taller and thicker, and then begins to merge at the center. At that point, the balance between the upward pulling Coulombic attraction and the laterally broadening Laplace pressure becomes critical. Depending on their relative strengths, the final state (marked as "F") can be either a nanocone or nanovolcanos with varying values of crater height. For example, Fig. 5c shows the simulation result obtained after the $\mu$ and $\gamma$ values changed to 100 cps and 0.08 N m$^{-1}$, respectively, which corresponds to the case of low-UV-dose and less-viscous NOA73. Even though the initial profile is identical to that in Fig. 5b, the final profile exhibits only a small dip at the center due to the dispersion and merging of the crater at the center. By iteratively adjusting the relative strengths of the Coulombic attraction and Laplace pressure in the simulation, we could reproduce the experimental results very closely. For instance, Fig. 5d shows the simulated surface height profile very closely agrees with those of the three nanovolcanos (Fig. 4i, Scan 1).

**Computational simulation of demolding-induced friction.** Given the supporting evidences for the ring charge formation due to the replica molding of PDMS nanocup replicated from PC nanocones, we sought the reason for such a spatially selective, non-uniform tribocharging. Our immediate hypothesis was that the PDMS nanocup's rim area sustained the highest level of friction during the demolding process which, in turn, increased the level of tribocharging in that region. To test the hypothesis, we carried out a nonlinear finite element analysis (FEA) of the cohesive demolding process. Details of the analysis setup and procedure can be found in Methods. The results are shown in Fig. 6.

Due to the spherical shape of the PDMS-PC interface, the detachment occurred in a 'mixed' mode, which combines the pure crack opening and the sliding modes. So, to compute $\sigma_f$, the frictional stress measured in Pa, we adopted the mixed mode cohesive zone model (CZM) in the presence of the nonlinearities both in material and geometry. Figure 6a–c shows that as the PDMS nanocup is gradually detached from the PC nanocone, the rim area experiences the maximum level of frictional stress. The complete temporal evolution of the frictional stress distribution along the progress of the demolding action is clearly visualized in Supplementary Movie 1.

To assess the cumulative impact of the frictional stress, we also computed the frictional fracture energy $G_f$, measured in J m$^{-2}$, by integrating the area under the frictional stress-tangential sliding curve over the whole process of demolding and plotted it in as a function of $L/L_{tot}$ in Fig. 6d, where $L$ and $L_{tot}$ are the arc lengths from the nanocup rim to the observation point and the nanocup bottom, respectively, as shown in the inset. It confirms that the cumulative frictional stress during the demolding process is concentrated near the rim, forming a peak covering up to $L \sim 0.2L_{tot}$, or ~60 nm in our nanocup setup, before decaying rapidly. It agrees well with our electrostatic modeling result which indicated that the surface charges formed a 55 nm-wide annular strip from the rim. Over the mid-to-bottom portion of the PDMS

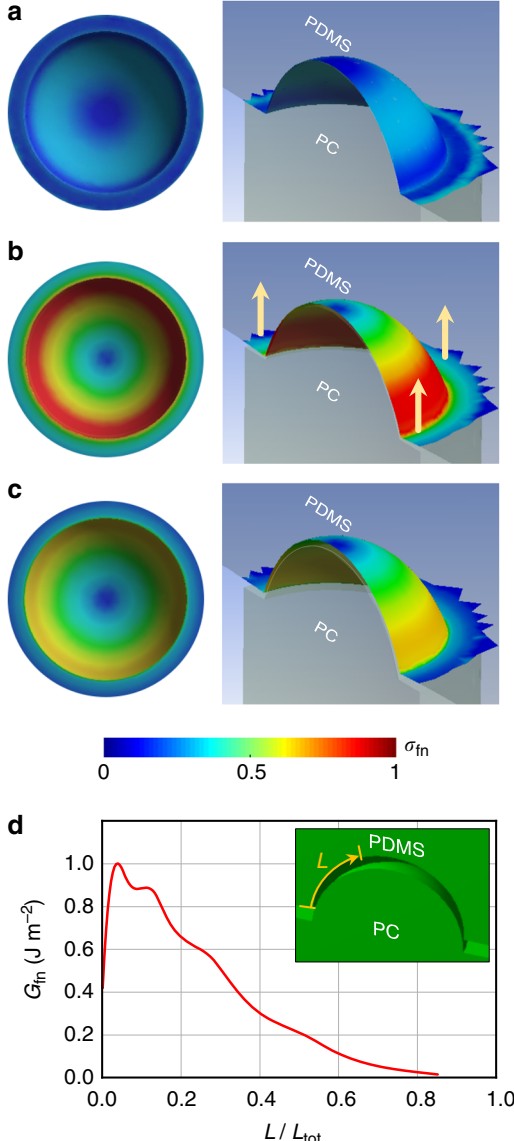

**Fig. 6** Computational analysis of demolding-induced friction. **a–c** The distribution of the frictional stress computed by nonlinear FEA. The left and right columns represent the top and cross-sectioned bird's eye views of a PDMS nanocup getting demolded from a PC nanocone, respectively. The color indicates $\sigma_{fn}$, the frictional stress normalized by its overall maximum. **a**, **b**, **c** Describe the PDMS nanocups in conformal contact with the PC nanocone, at the initial stage of the vertical demolding (along the direction indicated by the arrows), and at the starting point of the peel-off, respectively. The latter two clearly show that the demolding action induces the highest level of frictional stress around the nanocup's rim. **d** $G_{fn}$, the frictional fracture energy normalized to its maximum, as a function of $L/L_{tot}$. It further confirms that the frictional stress accumulated over the whole-demolding process maximizes around the rim. $G_{fn}$ exhibits a sharp peak up to $L \sim 0.2L_{tot}$ (~60 nm in our setup) before a rapid decay, in good agreement with our electrostatic modeling result, which indicated that the surface charges form a 55 nm-wide band from the rim

nanocup, the lack of intense frictional stress is likely to lead to a matching lack of tribocharging, rather than charging at the opposite polarity which will only weaken the EHDL efficiency. This analysis result not only gives further corroboration to our ring charge hypothesis but also provides useful insights for designing more elaborate replica molding-based tribocharge nanopatterning.

## Discussion

The technological contribution of this work is twofold. First, it introduced a technique to produce nanopatterned tribocharges on highly flexible PDMS surfaces capable of forming intimate contact with non-flat surfaces. It is a simple and effective technique which accomplishes both tribocharge generation and patterning in a single operation of replica molding. By generating the charge directly through triboelectrification, this technique also eliminates the need for external supply of electric charge, which often necessitates metallization of the elastomer surface.

Second, through the effort to utilize EHDL as a tool for tribocharge characterization, this work has also advanced the EHDL technique itself. In the conventional EHDL, which relies on electric fields generated by patterned electrodes, the polymer either forms an array of nanopillars under the electrode's surface pattern or simply mirrors the pattern itself through merging of the nanopillars, limiting the feature size to that of the electrode pattern or the characteristic length of the electrohydrodynamic instability[42–44,47]. Both are generally at micron-scales. Using the replica molding-induced nanopatterned tribocharges as the source of the electric fields, we have greatly reduced the EHDL's feature size. For instance, this work produced a highly regular array of submicron-scale nanovolcanos by decorating plain nanocones with 10 nm-scale nanocraters.

In addition, we have also integrated the EHDL process with a variety of spatially selective photopolymer texturing techniques, enabling it to realize multiscale nanotextures. For example, the nanovolcanos were monolithically integrated with microscale sinusoidal surface texture (~2.6 μm in pitch). More complex textures can be realized through a judicious choice of the UV-illumination pattern. To demonstrate the robustness of the tribocharge-enabled EHDL, we have also tried nanovolcano fabrication on NOA73 surfaces pre-textured in a totally different setup and geometry (Supplementary Fig. 3) and obtained affirmative results (Supplementary Fig. 4 and Supplementary Fig. 5), as detailed in Supplementary Note 1.

In conclusion, we have systematically investigated the intriguing phenomenon of replica molding-induced nanopatterning of tribocharge on elastomer surfaces using SKPM, electrohydrodynamic lithography, electrostatic modeling, and finite element analysis. Their results all pointed to the fact that the induced tribocharge's final distribution pattern is determined not only by the mold's surface topography but also by the mechanics of the demolding process. In particular, the level of frictional stress accumulated over the demolding action turns out to be of paramount importance. These findings provide deeper insights into nanoscale patterning of electric charges on elastomer surfaces. The technique will be especially useful for generating nanoscale annular charge distributions. With careful balancing of capillary action and Coulombic attraction, this triboelectrohydrodynamic lithography will become a versatile tool for fabricating functional materials and meta-surfaces.

## Methods

**Tribocharged nanocup array fabrication**. To fabricate the tribocharged PDMS nanocup array, we first prepared a PC mold with a 750 nm-pitch triangular array of nanocones (500 nm in base diameter, 150 nm in height, about $1 \times 1$ cm², Microcontinuum Inc.) and then poured liquid phase PDMS (Sylgard 184, Dow Corning) mixed with the curing agent at 10:1 wt. ratio. Upon its complete solidification, we peeled it off from the mold, obtaining a matching array of nanocups, 500 and 153 ± 13 (s.d.) nm in diameter and depth, respectively.

**AFM-based surface characterization and Kelvin probe microscopy**. The surface topography of the photopolymer after EHDL was examined by AFM in the tapping mode (Multimode, Veeco). To measure the surface topography and potential, we also configured the AFM in the SKPM mode. Antimony ($n$) doped Si tips (TESPA-V2, spring constant 42 N m$^{-1}$, resonance frequency of 320 kHz) and Pt/Ir coated tips (SCM-PIT, spring constant 2.8 N m$^{-1}$, resonance frequency of 75 kHz) were

purchased from Bruker. The lift height and the amplitude set-point were set to be 100 nm and 0.28 V, respectively.

**Electrostatic modeling of surface potential**. To compute the electric potential arising from the electric charges distributed on the nanocup's inner cavity surface, we first decomposed the inner cavity surface into a stack of thin annular strips with varying radii. Then we multiplied the preset surface charge density $\rho_s$ to the surface area of each annular strip to determine the corresponding total charge. We then modeled each annular strip as a ring charge distribution. The electric potential $V$ arising from a ring charge distribution with radius $a$ is given in closed form as[48]

$$V = \frac{Q}{2\pi^2 \epsilon_0} \cdot \frac{K\left(\sqrt{\frac{4a\rho}{a^2 + \rho^2 + h^2 + 2a\rho}}\right)}{\sqrt{a^2 + \rho^2 + h^2 + 2a\rho}} \qquad (4)$$

where $Q$, $\rho$, $h$, and $\epsilon_0$ are the total charge of the ring, the radial and vertical displacement of the observation point from the center of the ring, and the electric permittivity in vacuum, respectively. $K$ is the elliptic integral of the first kind. Then we summed up the contributions of the ring charges at each observation point. The number of the stacked rings was increased until the final summation converged.

**Sinusoidal pre-texturing of photopolymer**. The UV-curable photopolymer (NOA73, Norland Inc.) was spin-coated on the silicon substrate for 10 s at 500 r.p. m. and then 45 s at 3000 r.p.m., resulting in a thin film with thickness of ~40 μm. The photopolymer thin film was then exposed to a UV-two-beam interference pattern generated by the Lloyd mirror set-up employing a HeCd laser (Kimmon) installed on a floated optical table[49]. The pitch can be facilely controlled by the beam incident angle. The power intensity of the interference pattern on the photopolymer thin film was around 1 mW cm$^{-2}$ (power meter, 2931-C, Newport). The dose applied to the photopolymer was controlled by the exposure time, and hence the amplitude of the obtained one-dimensional surface relief structure can be accurately tuned. The AFM scans of two different types of sample sinusoidal textures on NOA73 are shown in Supplementary Fig. 2. Their profiles exhibit excellent agreements with the theoretically predicted sinusoidal pattern[49], signaling a successful two-beam interference. The strong crest-to-trough contrast, maintained even after several tens of minutes of exposure, also attests to the overall integrity of the Lloyd mirror setup.

**Tribocharge-enabled EHDL**. The flexible PDMS stamp with triangular nanocups array was placed in contact with the pre-structured photopolymer thin film for 2 min without applying any pressure, followed by additional UV exposure for 2 min (0.1 W cm$^{-2}$, Bluewave 200, Dymax) to fix the shape of the hierarchical nano-pattern. The PDMS stamp was then peeled off from the photopolymer thin film, resulting in the photopolymer multiscale nanotexturing on the silicon substrate.

**Numerical modeling of EHDL**. The EHDL process was computationally simulated by simultaneously solving the coupled Eqs. 2 and 3. Along the x-direction in Fig. 5a, the computational domain measured 4 μm and was discretized into 150 ~ 230 computation points. Along the h-direction, the extent was varied from its minimum at 100 nm, i.e., the gap between the PDMS replica and the NOA73 surfaces, depending on the shape of the charge distribution within the nanocup, which was modeled to exhibit an arc or a super-Gaussian profile. Since the simulation was carried out in 2D, the model charge distribution was configured to reproduce the 3D distribution pattern after revolution about the center axis. For example, a simple ring charge distribution was translated into two point charges located symmetrically about the center axis of the nanocup. More pairs were added to model charge distributions covering the nanocup's cavity wall. We computed the electric fields by applying Coulomb's law along the surface profile of the polymer and decomposing the result into components tangential and normal to the surface. Once the pressure term in Eq. 3 was evaluated, it was substituted into the right-hand side of Eq. 2 which, in turn, got integrated in time domain using Newton–Rahpson method. The integration time was set to 5.2 ps empirically. All computations were performed with Matlab (R2013b, Mathworks Inc.).

**Finite element analysis of demolding-induced frictional stress**. We performed a computational simulation to estimate the non-uniform distribution of the maximum frictional stress over the interface between the PDMS replica and the PC mold. Since the goal was to elucidate the spatiotemporal evolution of frictional stress on the spherical interface, we adopted the continuum-based nonlinear finite element analysis based on the cohesive zone model (CZM). All computational simulations were conducted on ANSYS (Release 18.2). We scaled up the nanocup structure to the micrometer length scale while preserving all the geometric features due to the length-scale limit of the continuum-based FEA program in ANSYS. The material and failure characteristics of the interface elements were modeled from literature[50–53]. In particular, the Young's modulus and the Poisson's ratio were set to 1.8 MPa and 0.45, respectively. The CZM was defined with 15 kPa for the normal and shear strengths and 330 μm for the separation limit. We assumed a clear interfacial failure without any fracture of PDMS fibrils, based on the

observation that the PC mold stayed usable and no PDMS fracture has been detected after repeated molding/demolding.

**Data availability**. The data supporting the plots and other findings of this study are available from the corresponding authors upon reasonable request.

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

## Acknowledgements

This work was supported by the National Science Foundation under grants CMMI-1265844 (J.K., A.P., Q.L., in part) and CBET-1605275 (J.K., Q.L., and I.C., in part), the Department of Civil, Construction, and Environmental Engineering of Iowa State University (I.C., in part), and the U.S. Department of Energy (DOE), Office of Science, Basic Energy Sciences, Materials Science and Engineering Division (R.B.). Ames Laboratory is operated for the U.S. DOE by Iowa State University under contract #DE-AC02-07CH11358. We thank Dr. Rabin Dhakal for valuable discussions.

## Author contributions

R.B. and J.K. conceived the idea and designed the experiments. Q.L. and A.P. carried out the experiments and data analysis. I.C. performed the finite element analysis. All authors participated in the interpretation of the data and writing of the manuscript.

## Additional information

**Competing interests:** The authors declare no competing interests.

