## [Peer Review File · Nature Communications]

Reviewers' comments:

Reviewer #2 (Remarks to the Author):

This paper describes a work on patterning non flat surfaces, using tribocharged PDMS and electrohydrodynamic lithography (EHDL).

Charge-patterning via tribocharging and the subsequent printing of nano-patterns of other objects (such as nanoparticles, e.g. Heiko Jacobs et al Adv. Mater. 2002,14,1553) have been extensively published in the past. The authors should cite some of these works in their manuscript. To the best of my knowledge, the first paper on the submicron charge patterning is Science 2001, 291, 1763, Heiko Jacobs and George Whitesides. Following this paper, there are many more, at least a couple of these should be cited here to inform the reader. (Also the title is a bit misleading, in my opinion - I guess the authors would like to emphasize the features they print via tribocharged surface and EHDL, not the observation of nanopatterned tribocharges, since the latter has already been shown before) Nevertheless, the presented work is novel in the sense that it combines (tribo)charge patterning with EHDL and that it creates patterns on non flat surfaces.

The manuscript is well written in terms of scientific detail.

_At one point the authors claim that the 'this work reveals the characteristics and mechanism of the nanoscale replica molding-induced tribocharging effect...' I think this statement is not right since the mechanism of tribocharging is not investigated here.

_Authors might also wish to comment on why they see a negative ring (Fig 1) rather than a mosaic of charges as in Ref. 9 (Baytekin et al Science 2011). Is the negative charge they report the 'net charge' on this particular region? (The mosaic of charges were reported by others, too:

Burgo T. A. L., et al. Triboelectricity: Macroscopic charge patterns formed by self-arraying ions on polymer surfaces. Langmuir 28, 7407-7416 (2012).

Knorr N. Squeezing out hydrated protons: Low-frictional-energy triboelectric insulator charging on a microscopic scale. AIP Adv. 1, 022119 (2011).

Pandey A., Kieres J., Noras M. A. Verification of non-contacting surface electric potential measurement model using contacting electrostatic voltmeter. J. Electrostat. 67, 453-456 (2009).

Barnes A. M., Dinsmore A. D. Heterogeneity of surface potential in contact electrification under ambient conditions: A comparison of pre- and post-contact states. J. Electrostat. 81, 76-81 (2016).

Terris B. D., Stern J. E., Rugar D., Mamin H. J. Contact electrification using force microscopy. Phys. Rev. Lett. 63, 2669-2672 (1989).)

_Can nanovolcanos form because the bottom of the PDMS well is differently polarized than the rim? Or it is not (tribo)charged at all upon peeling from the PC?

_Is it also possible to use positively charged PDMS to do a 'negative' printing?

While presented work would be an interesting read particularly for the triboelectricity and (nano)patterning communities, to my opinion, the paper falls short of presenting a broad-scope scientific work. Therefore I suggest the authors to have this detailed work published in a more specialized journal.

Reviewer #3 (Remarks to the Author):

In this work, the authors noticed the triboelectric charges on elastomer surfaces which can be used as a platform technique for multiscale surface texturing of polymers. However, the explanations in the manuscript is not convincing and the conjectures lacks experimental evidences. The iterative numerical simulation result is not enough to verify the conjectures. The authors should supply more experiments to explain the following questions clearly and directly.

1. The nanovolcano topography is conjectured to be caused by the ring tribocharge distribution in a

PDMS nanocup. But the generation of the ring charge distribution is not interpreted, nor experimentally verified. The explanation through iterative numerical simulation is indirect and unconvincing. On the other hand, the PDMS nanocups with different charge distributions (charge injected or depolarized at least) should be used as the contrast experiments.

2. The formation mechanism of the nanovolcano structure is still confusing. The authors should supply more experiments to reveal the influences of UV exposure dose, exposure time, sinusoidally texture structure, electric field distribution, etc.

3. The authors should explain why the NOA73 in the trough with higher viscosity resulted in higher nanocones. Besides, why is the depth of the sinusoidal textures with larger UV exposure dose in Figure 4i much smaller than that in Figure 4c and f?

4. More details about the experiments and simulations should be provided.

a) The charge distribution of PDMS in the simulation and the detailed simulation process.

b) The UV exposure dose of the crest portion of the sinusoidally texture.

c) The morphology of PDMS nanocups and the sinusoidally texture should be characterized.

Reviewer #2 (Remarks to the Author)

[1] This paper describes a work on patterning non flat surfaces, using tribocharged PDMS and electrohydrodynamic lithography (EHD). Charge-patterning via tribocharging and the subsequent printing of nano-patterns of other objects (such as nanoparticles, e.g. Heiko Jacobs et al Adv. Mater. 2002,14,1553) have been extensively published in the past. The authors should cite some of these works in their manuscript. To the best of my knowledge, the first paper on the submicron charge patterning is Science 2001, 291, 1763, Heiko Jacobs and George Whitesides. Following this paper, there are many more, at least a couple of these should be cited here to inform the reader.

- The authors thank the reviewer for the comment. In our original manuscript, we cited 2 papers more recently published by Jacobs and coworkers (as Ref. [8] and Ref. [11]) but failed to trace the work down to their common root. In the revision, we cited the *Science* (2001) and *Advanced Materials* (2002) papers as the pioneering work in the revision.

[2] Also the title is a bit misleading, in my opinion - I guess the authors would like to emphasize the features they print via tribocharged surface and EHD, not the observation of nanopatterned tribocharges, since the latter has already been shown before.

- The authors agree that the original title "*Observation of nanopatterned triboelectric charges on elastomer surfaces induced by replica molding*" was misleading in emphasizing only the nanopatterned tribocharge generation while ignoring its utilization in EHD-based nanotexturing.
- In the revision, we changed the title to "*Replica molding-based nanopatterning of tribocharge on elastomer with application to electrohydrodynamic nanolithography*" to emphasize both the charge patterning and EHD-based texturing aspects of our work.
- The tribocharge nanopatterning aspect was not de-emphasized in the new title since we still consider it novel. Multiple prior reports exist on nanopatterning of tribocharge but we believe that ours is the first to explicitly link it to the process of replica molding and, more importantly, to the frictional stress pattern arising from the demolding process. In the revision, this connection becomes much stronger thanks to our new, more detailed analysis of the surface Kelvin probe microscopy results and nonlinear finite element analysis (FEA) of the cohesive demolding process.
- Since the replica molding is the central and essential enabling factor of our work, we used the expression "*Replica molding-based*" as the entry words of the new title.
- In addition, we totally re-wrote the subsection of "Discussion" in the revision to elaborate on the uniqueness of our technique.

[3] Nevertheless, the presented work is novel in the sense that it combines (tribo)charge patterning with EHDL and that it creates patterns on non flat surfaces. The manuscript is well written in terms of scientific detail.

- The authors appreciate the encouraging comments.

[4] At one point the authors claim that the 'this work reveals the characteristics and mechanism of the nanoscale replica molding-induced tribocharging effect...' I think this statement is not right since the mechanism of tribocharging is not investigated here.

- The authors agree that it was an overstatement and removed it from the revision.
- In parallel, the authors would like to assure the reviewer that the revised manuscript contains substantially more results obtained from our extended experimental and theoretical efforts to elucidate the tribocharge nanopatterning mechanism.
- They appear in “Electrostatic modeling of surface potential (pg. 4)”, “Computational simulation of demolding-induced friction (pg. 12)”, and “Discussion (pg. 14).” The new plots and figures in Fig. 3 and Fig. 6 in the revision also describe the new results in detail.

[5] Authors might also wish to comment on why they see a negative ring (Fig 1) rather than a mosaic of charges as in Ref. 9 (Baytekin et al Science 2011). Is the negative charge they report the 'net charge' on this particular region?

- The authors thank the reviewer for pointing that out. We indeed think that the tribocharge can be in mosaic form with the net charge determining the overall polarity. The original manuscript did not comment on it explicitly since it lacked quantitative analysis of the induced tribocharge density.
- We have performed new, more detailed analysis of the surface Kelvin probe microscopy scans (revised Fig. 3e) that reveal unusual surface potential exhibiting peaks near the center of the nanocups. This yields valuable information on the ring-charge distribution within the nanocups
- The SKPM measurements were then paired with the 3D electrostatic modeling which enabled direct estimation of the net charge density in this revision. The details are given in the subsections of “Electrostatic modeling of surface potential (pg.4)” and the corresponding part of Methods.
- The result indicates that the net charge density was ~ 0.6 net negative elementary charges per 10 nm^2 which is in order-of-magnitude agreement with the result reported by Baytekin *et al.* (1 net negative elementary charge per 10 nm^2).

[6] The mosaic of charges were reported by others, too:

Burgo T. A. L., et al. Triboelectricity: Macroscopic charge patterns formed by self-arraying ions on polymer surfaces. *Langmuir* 28, 7407-7416 (2012).

Knorr N. Squeezing out hydrated protons: Low-frictional-energy triboelectric insulator charging on a microscopic scale. *AIP Adv.* 1, 022119 (2011).

Pandey A., Kieres J., Noras M. A. Verification of non-contacting surface electric potential measurement model using contacting electrostatic voltmeter. *J. Electrostat.* 67, 453-456 (2009).

Barnes A. M., Dinsmore A. D. Heterogeneity of surface potential in contact electrification under ambient conditions: A comparison of pre- and post-contact states. *J. Electrostat.* 81, 76-81 (2016).

Terris B. D., Stern J. E., Rugar D., Mamin H. J. Contact electrification using force microscopy. *Phys. Rev. Lett.* 63, 2669-2672 (1989).

- The authors thank the reviewer for the information. They are all cited in the subsection of “Electrostatic modeling of surface potential (pg. 4)” of the revision in conjunction with the estimation of the net surface charge density.

[7] Can nanovolcanos form because the bottom of the PDMS well is differently polarized than the rim? Or it is not (tribo)charged at all upon peeling from the PC?

- We hypothesize that the nanovolcano was formed through an EHDL induced by a ring charge, *i.e.*, negative charges concentrated only around the nanocup’s rim, leaving the mid-to-bottom portion of the PDMS nanocup’s inner surface uncharged (pg. 13).
- In the original manuscript, the hypothesis was tested only indirectly through numerical reconstruction of the EHDL results. Regarding this, Reviewer #3 also expressed doubts on the ring charge model’s validity. So we carried out additional analyses.
- First, we re-examined the surface Kelvin probe microscopy scan results along different directions and revealed the unique “peak-in-the-well” potential profile (Fig. 3c of the revision) which can be specifically traced down to ring-shaped charge distributions. Similar potential profiles have been also observed from the ring charges deliberately prepared by Jacobs *et al.* (*Science*, 2001). The fact that the measured potential throughout the whole cavity stayed well below the potential level along the nanocup’s rim indicates that the portion of the PDMS nanocup not charged negatively is uncharged or positively charged at a negligible density. Either case validates our negative ring charge model. This argument is added to the “Electrostatic modeling of surface potential” subsection of the revision.

- Second, we also numerically simulated the generation of frictional stress during the “peeling-off” process and confirmed that the mid-to-bottom area sustains much weaker frictional stress than that of the rim area. Such a lack of friction is likely to result in a matching lack of tribocharging rather than charging with opposite polarity. This argument is added to the “Computational simulation of demolding-induced friction (pg. 12).”

[8] Is it also possible to use positively charged PDMS to do a 'negative' printing?

- In our EHD, the target polymer, NOA73, is originally neutral. The source charges polarize the polymer oppositely and then induce the Coulomb attraction force. So, we think that regardless of the source charge’s polarity the overall force stays attractive. Co-existence of both positive and negative charges at close proximity will only weaken the attraction.
- The argument above is added to “Computational simulation of demolding-induced friction (pg. 12)” subsection.

[9] While presented work would be an interesting read particularly for the triboelectricity and (nano)patterning communities, to my opinion, the paper falls short of presenting a broad-scope scientific work. Therefore I suggest the authors to have this detailed work published in a more specialized journal.

- The authors respectfully ask for the reviewer’s reconsideration on the basis of the highly interdisciplinary nature of our work. We specifically chose *Nature Communications* as the potential vehicle to disseminate our results due mainly to its openly announced advocacy of interdisciplinary research and broad readership.
- Our work is truly interdisciplinary and multi-physical. The “interesting” results were obtained by fusing many seemingly unrelated aspects gleaned from
 - Optical (use of photopolymer and its two-beam interference-based preparation),
 - Electrical (use of surface charge distributions and the resulting electric fields),
 - Mechanical (use of dynamic friction and rheological EHD effects)

fields of study, involving researchers from Physics, Electrical Engineering, and Civil Engineering.

- Once published, our work will broadly interest and benefit diverse scientific communities studying nanofabrication, nanomanufacturing, material science, electronics, photonics, and bio-MEMS/NEMS.
- We are concerned that reporting this result through a specialized journal would only lead to a short-term, localized impact. The authors sincerely ask for the reviewer’s reconsideration.

Reviewer #3 (Remarks to the Author)

In this work, the authors noticed the triboelectric charges on elastomer surfaces which can be used as a platform technique for multiscale surface texturing of polymers. However, the explanations in the manuscript is not convincing and the conjectures lacks experimental evidences. The iterative numerical simulation result is not enough to verify the conjectures. The authors should supply more experiments to explain the following questions clearly and directly.

[1] The nanovolcano topography is conjectured to be caused by the ring tribocharge distribution in a PDMS nanocup. But the generation of the ring charge distribution is not interpreted, nor experimentally verified. The explanation through iterative numerical simulation is indirect and unconvincing. On the other hand, the PDMS nanocups with different charge distributions (charge injected or depolarized at least) should be used as the contrast experiments.

- The authors agree that the supporting evidence in the original manuscript was indirect and limited. To strengthen our work, we have performed the following additional experiments and analyses.
- First, we analyzed the existing scanning Kelvin probe microscopy (SKPM) measurements (shown as Figs. 3b and c of the original manuscript) along different scan directions. Figures 3b and c are re-plotted in the revision with the data extraction direction changed to emphasize the appearance of the center peaks in the middle of the nanocup's potential wells. According to our numerical analysis, described in detail in the newly added Fig. 3e and in "Electrostatic modeling of surface potential (pg. 4)" subsection, such a "peak-in-the-well" potential profile is a signature characteristic of the existence of a ring charge, rather than a uniform dome of charge. Figure 3e also shows that without the ring charge distribution, the "peak-in-the-well" will be much lower, leaving us unable to explain the experimental SKPM result.
- Second, a "dip-on-the-peak" potential profile, the inverse of our "peak-in-the-well" profile, has also been observed from the positive ring charges deliberately prepared by Jacobs *et al.* (Ref. [7] of the revision).
- Third, we also numerically simulated the generation of frictional stress during the "peeling-off" process and confirmed that the mid-to-bottom area sustains much weaker frictional stress than the rim area does. Such a lack of friction is very likely to result in a matching lack of tribocharging, casting the charge into a ring form. This argument is added to the "Computational simulation of demolding-induced friction" subsection (pg.12).
- In the original manuscript, we supported our ring charge hypothesis only through the reconstruction of the EHD result. In the revision, we provide three more supporting evidences and one of them is through direct measurements. The authors wish that these

supporting evidences could reinforce our validation of the ring charge hypothesis and convince the reviewer.

- At the reviewer's suggestion, we tried depolarization of the tribocharged PDMS nanocups by immersing them in DI water or heating them with hot plate. Subsequent SKPM indicated that the charge distribution was certainly disturbed but did not disappear entirely. Beyond that, we could not find a good way to depolarize the sample.

[2] The formation mechanism of the nanovolcano structure is still confusing. The authors should supply more experiments to reveal the influences of UV exposure dose, exposure time, sinusoidally texture structure, electric filed distribution, etc.

- The task is very challenging since most factors, especially the dose, exposure time, and texture distribution, are inter-related, making it difficult to vary one independently of another. Therefore we attempted to back up the basic principle of tribocharge-enabled EHDL, and also its robustness, by repeating the process in totally different setups and checking if the nanovolcano could still be formed.
- Specifically, we tried to induce the nanovolcano formation on a NOA73 surface pre-textured through replica molding, rather than the two-beam interference. Accordingly, the pitch of the surface texture was also changed from the original 2.6 μm to 1.7 μm and 5.0 μm . The pre-texture morphology has also been changed from the ~ 50 nm-deep sinusoidal profile to ~ 60 nm-deep trench profile. Using these disparate setups, we tested whether

(1) The tribocharge-enabled EHDL works,

(2) The viscosity-controlled switching between nanocone and nanovolcano works.

- The results are summarized in Supplementary Note 1 with its figures in Supplementary Figures 3, 4, and 5. A lead sentence was also inserted into the 3rd paragraph of "Discussion" (pg. 14)
- To summarize, both (1) and (2) got checked out successfully. Higher viscosity did lead to the formation of nanovolcanos in the trough despite a variety of changes made to the setup. The results re-affirm the validity and robustness of the nanovolcano formation mechanism suggested in our work.

[3] The authors should explain why the NOA73 in the trough with higher viscosity resulted in higher nanocones.

- The authors are delighted that the intriguing phenomenon is pointed out by the reviewer. In fact, we regard it as an important piece of evidence supporting the ring charge hypothesis.
- Accordingly, we had attempted to explain the reason in the original manuscript's "Results" section but the explanation happened to be spread over two different subsections.
- In the revision, for more coherent explanation, we combined the explanations and put them into a new subsection titled "Underfilled crest nanocone as ring charge evidence." Currently, it is in page 8 of the revised manuscript.
- To summarize, as the title of the subsection implies, we think that the crest nanocones are formed by capillary filling of the PDMS nanocup. The ring charges around the nanocup's rim, however, countered the capillary action through Coulombic attraction and frustrated the growth of the nanocone. However, as explained in "Nanovolcano formation as ring charge evidence" (pg. 9), the trough nanocones and nanovolcanos are formed through EHD which is aided, rather than frustrated, by the Coulombic attraction. In consequence, the structures in the trough become higher despite higher viscosity.

[4] Besides, why is the depth of the sinusoidal textures with larger UV exposure dose in Figure 4i much smaller than that in Figure 4c and f?

- It can be explained by the fact that both EHD and capillary filling require additional photopolymer to sustain their upward deformation. In other words, the nanocones in the trough in Fig. 4c achieved their height by lowering the bottom level around them, generating a false impression of a deeper trough in the process.
- This explanation is added to the "Nanovolcano formation as ring charge evidence" subsection in "Results" of the revision (pgs. 9,10).

[5] More details about the experiments and simulations should be provided.

[5a] The charge distribution of PDMS in the simulation and the detailed simulation process.

- The authors thank the reviewer for pointing out the obvious deficiency in the manuscript. Accordingly, we added a new subsection titled "Numerical modeling of EHD" in "Methods" of the revision (pg. 18).

[5b] The UV exposure dose of the crest portion of the sinusoidally texture.

- The intensity pattern of the Lloyd mirror-produced 2-beam interference takes the form of a sine squared function. Therefore, ideally the crest portion is supposed to have received little or no dose.
- The AFM scans of the sinusoidal textures, shown in Supplementary Fig. 2 in the Supplementary Figures in response also to [5c], indeed exhibit excellent sinusoidal profiles, indicating that the operational principle of the Lloyd mirror-based 2-beam interference, including the “zero-dose at the null”, was faithfully accomplished.
- One major factor that may lead to a significant degradation of the ideal interference is the mechanical vibration of the setup which “washes out” the interference pattern. In our setup which was used to obtain the results in Supplementary Fig. 2, the optical table was floated and the NOA73 film was exposed for 60~80 minutes. If there were any vibration, the interference pattern could not have survived.
- With all these observations, the authors would like to assure the reviewer that the interference pattern was nearly ideal and the crest portion received little or no dose.
- Our new statements are included in the revision’s “Sinusoidal pre-texturing of photopolymer” subsection of “Methods” as (pg. 17):

Their profiles exhibit excellent agreements with the theoretically predicted sinusoidal pattern, signaling a successful two-beam interference. The strong crest-to-trough contrast, maintained even after several tens of minutes of exposure, also attests to the overall integrity of the Lloyd mirror setup.

[5c] The morphology of PDMS nanocups and the sinusoidally texture should be characterized.

- The morphology of the PDMS nanocup array is characterized in more detail through scanning electron microscopy and the result is included as Supplementary Fig. 1 in Supplementary Figures. It is referenced in the “Kelvin probe microscopy of nanopatterned tribocharge” subsection of “Results” (pg. 4).
- We have also included new atomic force micrographs of sinusoidal textures induced by UV 2-beam interference on NOA73 as Supplementary Fig. 2 in Supplementary Figures. It is referenced in the “Sinusoidal pre-texturing of photopolymer” subsection of “Methods” (pg. 17).

REVIEWERS' COMMENTS:

Reviewer #3 (Remarks to the Author):

In the revised manuscript, the authors supplied some significant experiments and analyses. The simulation of demolding-induced friction provided powerful supporting evidence for the generation and distribution of the ring charge. It is convincing that the tribo-charges are located in the stronger friction area. The manuscript can be published after the authors reconsider the following questions.

1. The authors stated that the crest portion of the sinusoidal texture received little or no dose. If there is no dose in the crest portion in Figure 4h (line 2, red), there will be no nanovolcano observed. The curves (Figure 4i) should be similar to that in Supplementary Figure 5 (nanocones in the crest and nanovolcanos in the trough). The two figures should be consisted with each other.
2. The authors are suggested to refine their expressions in the introduction and discussion sections.

Reviewer #3 (Remarks to the Author)

In the revised manuscript, the authors supplied some significant experiments and analyses. The simulation of demolding-induced friction provided powerful supporting evidence for the generation and distribution of the ring charge. It is convincing that the tribo-charges are located in the stronger friction area. The manuscript can be published after the authors reconsider the following questions.

[1] The authors stated that the crest portion of the sinusoidal texture received little or no dose. If there is no dose in the crest portion in Figure 4h (line 2, red), there will be no nanovolcano observed. The curves (Figure 4i) should be similar to that in Supplementary Figure 5 (nanocones in the crest and nanovolcanos in the trough). The two figures should be consisted with each other.

- The authors agree that the results in Fig. 4i and Supplementary Fig. 5 (and also 4) are different from each other.
- Regarding the issue, we would like to point out that the two sets of result were obtained under different setups, exposure dose levels, and experimental conditions.
 - The crest/trough patterns in Fig. 4i were obtained with 2-beam interference of a coherent UV laser with the dose set at 3.6 J/cm^2 .
 - The crest/trough patterns in Supplementary Figs. 4 and 5 were obtained through replica molding and partial curing by an incoherent UV source with the dose set at a lower value of $1.3\sim 2.1 \text{ J/cm}^2$.
- Such differences in the experimental setups may have led to the following consequences.
 - In the case of Fig. 4i, it is likely that the crest region was exposed to UV dose to an appreciable level since the exposure time was very long (~ 60 minutes). Even though the optical table was floated, it was impossible to eliminate the vibration entirely. Such a non-zero vibration leads to incomplete destructive interference and residual UV dose in the crest region. We think that the residual UV dose has reduced the fluidity in the crest NOA73. Since the crest area is in contact with the PDMS nanocups, the main mechanism for its shaping is capillary action. The reduction in the fluidicity, in combination with the Coulombic attraction from the rim charges mentioned in the “Underfilled crest nanocone as ring charge evidence” subsection of the main text, could impede the complete filling of the nanocup and the formation of perfect nanocones, leaving them with small dips as shown in Fig. 4i. Note that the dips are noticeably shallower than the craters in the trough, indicating that the two substructures were originated from different mechanisms.

- On the other hand in supplementary Fig. 5, the crest portion of the long-period corrugations of NOA73 received lower dose and could stay more fluidic and deformable before the EHDL process, which led to the formation of only nanocones.
- To clarify these points, we appended the following sentences to the Supplementary Note:

It is worth noting that in the two-beam interference-based EHDL, nanovolcanos were formed even in the crest region as shown in Fig. 4i (red trace). Although their nanocraters are lower than those in the trough region, such a nanovolcano formation contrasts the results in Supplementary Figs. 4 and 5 in which the crest portion is occupied only by nanocones. We ascribe the discrepancy to a possible incomplete destructive interference. Even though the optical table was floated, the long exposure time (~60 minutes) may have imparted residual UV dose on the crest region, rendering the area more viscous and less deformable. Since the crest area is in contact with the PDMS nanocup, the main mechanism for its shaping is capillary action. The reduction in the fluidicity, in combination with the Coulombic attraction from the rim charges mentioned in the “Underfilled crest nanocone as ring charge evidence” subsection of the main text, could impede the complete filling of the nanocup and the formation of perfect nanocones, leaving them with small dips as shown in Fig. 4i. The higher UV dose (~3.6 J·cm² as opposed to 1.3~2.1 J·cm² in Supplementary Figs. 4 and 5) could have aggravated the process. Eventually, they could lead to an incomplete nanocone formation, i.e., the appearance of the shallow dip at the center.

[2] The authors are suggested to refine their expressions in the introduction and discussion sections.

- The authors appreciate the suggestion. In accordance, we thoroughly re-read the two sections and refined the texts. The changes are set in red font in the revision.